# Maternal Vaginal Colonization and Extended-Spectrum Beta-Lactamase-Producing Bacteria in Vietnamese Pregnant Women

**DOI:** 10.3390/antibiotics10050572

**Published:** 2021-05-13

**Authors:** Nguyen Thanh Viet, Vu Van Du, Nghiem Duc Thuan, Hoang Van Tong, Nguyen Linh Toan, Can Van Mao, Nguyen Van Tuan, Srinivas Reddy Pallerla, Dennis Nurjadi, Thirumalaisamy P. Velavan, Ho Anh Son

**Affiliations:** 1Institute of Biomedicine and Pharmacy, Vietnam Military Medical University, Hanoi 121-08, Vietnam; nguyenthanhviet@vmmu.edu.vn (N.T.V.); hoangvantong@vmmu.edu.vn (H.V.T.); 2National Hospital of Obstetrics and Gynecology, Hanoi 110-02, Vietnam; dutruongson@gmail.com; 3ENT Department, 103 Military Hospital, Vietnamese Military Medical University, Hanoi 121-08, Vietnam; thuanbm6@gmail.com; 4Department Post-Graduate Training Management, Vietnamese Military Medical University, Hanoi 121-08, Vietnam; toannl@vmmu.edu.vn; 5Department of Pathophysiology, Vietnamese Military Medical University, Hanoi 121-08, Vietnam; canvanmao2011@gmail.com; 6Department of Rehabilitation, Vietnamese Military Medical University, Hanoi 121-08, Vietnam; nguyentuan.351975@gmail.com; 7Institute of Tropical Medicine, Universitätsklinikum Tübingen, Wilhelmstrasse 27, 72074 Tübingen, Germany; 8Department of Infectious Diseases, Medical Microbiology and Hygiene, University Hospital Heidelberg, Im Neuenheimer Feld 324, 69120 Heidelberg, Germany; dennis.nurjadi@uni-heidelberg.de; 9Vietnamese-German Centre for Medical Research (VG-CARE), Hanoi 116-10, Vietnam

**Keywords:** pregnant women, Enterobacterales, antimicrobial resistance, Vietnam, extended-spectrum beta-lactamase, *Escherichia coli*, *Klebsiella pneumoniae*, carbapenem resistance

## Abstract

Extended-spectrum β-lactamase-producing Enterobacterales (ESBL-E) resistance to commonly prescribed drugs is increasing in Vietnam. During pregnancy, ESBL-E may predispose women to reproductive tract infections and increases the risk for neonatal morbidity. Vaginal colonization and infections by *Escherichia coli* and *Klebsiella pneumoniae* are seldom studied in Vietnam. In this study, we investigated ESBL-producing Enterobacterales in the birth canal of pregnant women. Between 2016 and 2020, vaginal swabs were collected from 3104 pregnant women (mean gestational age of 31 weeks) and inoculated onto MacConkey agar plates. Colonies were subjected to direct identification and antimicrobial susceptibility testing using the VITEK^®^-2 automated compact system and disk diffusion. ESBL production was determined phenotypically. *E. coli*, *Klebsiella species* were identified in 30% (918/3104) of the vaginal swabs, with *E. coli* being the most common (73%; 667/918). ESBL-production was detected in 47% (432/918) of Enterobacterales, with frequent multidrug-resistant phenotype. The overall prevalence of carbapenem resistance was low (8%). Over 20% of *Klebsiella* spp. were carbapenem-resistant. Pregnant women had a high prevalence of colonization and may transmit ESBL-E to neonates at birth, an important risk factor to be considered. The high rate of ESBL-producers and carbapenem resistance in Enterobacterales in Vietnam emphasizes the need for consequent surveillance and access to molecular typing.

## 1. Introduction

Antimicrobial resistance (AMR) is increasingly recognized as a global public health issue in many countries. With a population of more than 96 million and a high burden of infectious diseases, Vietnam faces a significant increase in AMR in the last years.

In low and middle-income countries, antibiotic resistance is increasing with high morbidity and mortality, but resistance data has been poorly documented. The WHO recently declared that the group of bacteria most commonly causing hospital-acquired infections (HAIs) in Asia as priority 1 includes *Escherichia coli* and *Klebsiella pneumoniae* [1]. Extended-spectrum β-lactamases producing Enterobacterales (ESBL-E) infections are increasing in Asia, with *E.*
*coli* and *K. pneumoniae* being the predominant contributors of ESBL-E in Vietnam [2].

Reports from 2002 until 2011 suggest an exponential increase in resistance due to ESBLs ESBL from 39 to 55% in Southeast Asia [3]. In particular, the Philippines and Vietnam have reported a high burden of ESBL-producing *E. coli* infections of 59% and 81%, respectively [4]. A point prevalence survey from Vietnamese pediatric hospitals shows 30% of patients from Vietnamese intensive care units across 16 hospitals had HAIs [5]. In addition, evident that Vietnamese neonatal patients with culture-confirmed Gram-negatives from HAIs had a 50% case fatality rate [6].

Maternal colonization with group B streptococci (GBS) and Enterobacterales is considered a significant risk factor for the acquisition of neonatal sepsis [7]. While AMR is not a major problem for GBS [8], resistance to beta-lactams in Enterobacterales is emerging and poses a clinical challenge. Among all antibiotic substances, beta-lactams are considered a safe option to treat infections during pregnancy, backed with good-quality evidence [9]. Colonized mothers may transmit ESBL-E to neonates at birth. Thus, colonization by ESBL-producers is an important risk factor in pregnant women [10]. Colonization rates of ESBL-E in pregnant women vary from 3–15% [10,11,12]. In particular, maternal vaginal colonization has repeatedly been demonstrated as a risk factor for transmission and hence can cause neonatal sepsis [7,9,13]. The emergence of ESBL-producers, especially in *E. coli* and *K. pneumoniae* as vaginal colonizers is alarming and of clinical relevance due to limited antibiotic options to treat pregnant women and neonatal early onset sepsis [14].

In Vietnam, the incidence of ESBL-producing strains among clinical *E. coli* and *Klebsiella* spp. isolates in vaginal swabs of pregnant women have not been studied. The knowledge of the burden of colonization in pregnant women and the extent of maternal-neonatal transmission is important for AMR surveillance and subsequent prevention strategies [15]. In this context, this study aimed to determine the incidence of ESBL-E in pregnant women attending the national hospital in the last five years (2016–2020).

## 2. Results

### 2.1. Baseline Characteristics of Enrolled Participants

A total of 3863 pregnant women with a mean age of 30 years (17–46) and a mean gestational age of 31 weeks (range 16–41 weeks) were enrolled in the study. Vaginal swab specimens were collected from 3104 pregnant women and were subsequently investigated (Table 1).

### 2.2. ESBL-E Carriage in Pregnant Women

A total of 3104 vaginal swabs from pregnant women were analyzed, and 918 (30%) isolates were positive for *E. coli* and *Klebsiella* spp. Among them, *E. coli* was predominant (*n* = 667/3104; 21%), followed by *K. pneumoniae* (*n* = 243/3104; 8%), and *K. oxytoca* (*n* = 8/3104; 0.3%). The rates of colonization by ESBL-positive and ESBL-negative Enterobacterales in all swab samples were 14% (*n* = 432) and 16% (*n* = 486), respectively. ESBL production was detected in 47% (432/918). Among the 432 ESBL-positive isolates, *E. coli* was predominant (*n* = 340/432; 79%), followed by *K. pneumoniae* (*n* = 88/432; 20%), and *K. oxytoca* (*n* = 4/432; 1%). There were no significant differences in the distribution of *E. coli* and *Klebsiella* spp. between the ESBL-positive and negative isolates (Table 2).

### 2.3. ESBL-E and Antibiotic Resistance

The ESBL-E isolates identified in this study showed antibiotic resistance to different classes of antibiotics, as summarized in Table 3. Of the 918 ESBL-Enterobacterales isolates, mostly from *E. coli* and *Klebsiella* spp., 97% (*n* = 891/918) showed antibiotic resistance to at least one antimicrobial agent. Due to nonuniformity in the availability of susceptibility data, only resistance data with complete antimicrobial susceptibility for ampicillin, ceftriaxone, ertapenem, gentamicin, ciprofloxacin and trimethoprim/sulfamethoxazole (*n* = 904/918) were considered for better comparability (Table 3). In general, the ESBL-positive isolates showed higher antibiotic resistance compared to the ESBL-negative ones. Only 8% (72/904) were phenotypically resistant to carbapenem. Resistance to multiple antibiotics was common; the prevalence of multidrug resistance (MDR) was higher in ESBL-positive isolates (65% vs. 39%, *p* < 0.001, Table 3)

Of the 667 *E. coli* isolated from the vaginal swabs, 51% (340/667) were ESBL producers. MDR was frequent in ESBL-positive *E. coli* than ESBL-negative *E. coli* (70% vs. 42%, *p* = 0.001). Co-resistances to gentamicin, ciprofloxacin and trimethoprim/sulfamethoxazole were frequent. Only 3% (22/657) of *E. coli* tested were resistant to carbapenem (Table 3).

Over 50% of *Klebsiella* spp. collected from the vaginal swabs were beta-lactam resistant. Of 243 *K. pneumoniae*, 36% (88/243) and of 8 *K. oxytoca*, 50% (4/8) were ESBL-producers. Like *E. coli*, MDR was more frequent in ESBL-producers (49% vs. 32%, p = 0.008). In ESBL-producers, the prevalence of carbapenem resistance was low (3% towards ertapenem and 9% towards imipenem). Surprisingly, a significantly higher percentage of ESBL-negative *Klebsiella* spp. were carbapenem-resistant (30% towards ertapenem and 33% towards imipenem, Table 3).

## 3. Discussion

Maternal vaginal colonization is an important risk factor in pregnant women, and ESBL-producing Enterobacterales can transmit pathogens to the newborns [16]. There is limited information on reproductive tract infections in resource-poor settings, such as Vietnam. To the best of our knowledge, this is the first study, which aimed to investigate the incidence of ESBL-producing Enterobacterales and to understand antibiotic resistance to commonly prescribed drugs.

In the present study, the prevalence of vaginal Enterobacterales and ESBL-Enterobacterales colonization among pregnant women was 30% and 10%, respectively. The ESBL-E incidence observed in this study is higher than other upper-middle-income countries, such as Norway (3%) [15] and Argentina (5%) [17], but lower than in other African countries (17%), respectively [18]. Nonuniform approaches to study design, particularly differences in inclusion criteria, age groups, the inclusion of hospital and/or community populations, etc., may modulate the reported ESBL incidences across geographic regions [19]. In addition, using standardized and harmonized AMR sampling, detection and interpretation procedures may additionally contribute to the differences in ESBL-E incidences.

In this study, *E. coli* and *K. pneumoniae* are the most commonly isolated gram-negative Enterobacterales and were known to be the major causative agents of neonatal ESBL-E infections [20,21]. Furthermore, these strains are reported to be the predominant uropathogenic bacteria in pregnant women in developing countries in Africa and Asia [22] and are acknowledged as common causes of community and hospital-acquired bacteremia in adults in Vietnam [23,24]. The Global Antimicrobial Surveillance System (GLASS) lists *E. coli* and *K. pneumoniae* as the two most common resistant priority pathogens [1,25]. The vaginal *E. coli* colonization rate of 21% observed in this study is a risk factor for preterm birth [26]. Vaginal *E. coli* colonization is associated with obstetric infections and neonatal infections. The colonization rate was higher than other studies, e.g., 20% in Lithuania [27], 14% in Argentina [28]. The 8% rate of *K. pneumoniae* isolates observed in this study is lower than those reported from Bangladesh (9%) [29] and in Nigeria (15%) [30]. The rates of ESBL-E producing *E. coli* and *K. pneumoniae* vary depending on the regions [31] and ESBL producing *E. coli* in this study was 11%, which is significantly higher [28].

In the presented study, the Enterobacterales isolates were observed to be more resistant to commonly prescribed drugs, such as ciprofloxacin (41%), trimethoprim/sulfamethoxazole (69%), and third-generation cephalosporins (56%, suggesting that these drugs may not be effective in treating infections in pregnant women. Ampicillin is the first-line antibiotic for treating obstetric and neonatal infections [32], and resistance to ampicillin is alarming for *E. coli*. A recent review showed a high prevalence of antibiotic resistance to commonly used antibiotics in commensal *E. coli* isolates from humans in low-middle-income countries (LMICs), including Vietnam; ampicillin (72%), oxytetracycline (78%), tetracycline (67%), streptomycin (58%) and trimethoprim (67%), chloramphenicol (45%), ciprofloxacin (17%), co-trimoxazole (63), nalidixic acid (30%) and third-generation cephalosporin–cefotaxime (27%) [33]. In this study, *E. coli* resistance to ampicillin was 92% and 56% to third-generation cephalosporin, higher than another study [28]. Furthermore, over 50% of *Klebsiella* sp. are resistant to third-generation cephalosporin, and over 36% were ESBL-producers, which is in line with previously reported epidemiological data from this region [34]. Unexpectedly, our data indicated that over 20% of *K. pneumoniae* and *K. oxytoca* were carbapenem-resistant, which was much higher than anticipated. The acquisition of carbapenem-resistant Enterobacterales in Vietnam through contact with hospital settings has been described previously [35]. The high rate of carbapenem resistance in *Klebsiella* spp. is alarming and warrants further investigations.

Most of the commonly prescribed drugs revealed high resistance; drugs such as broad-range penicillin, cephalosporins and carbapenems, may be considered as an alternative treatment option. Carbapenems are considered drugs of choice for infections caused by ESBL-producing enterobacteria [36,37]. However, the increased use of carbapenems for ESBL—*K. pneumoniae* infections has led to the rapid emergence of carbapenem resistance [38]. Therefore, using carbapenems as empiric antimicrobial therapy should be prudent.

There are a few study limitations. A major limitation of this study is that AMR data were generated in the routine microbiological diagnostic and molecular characterization could not be performed routinely due to limited resources. Since bacterial isolates were not cryopreserved, retrospective genotypic analysis for ESBL-producers could not be performed. Nonetheless, the presented findings highlight the ongoing AMR problem in Vietnam and raise important issues to access molecular typing methods/facilities in low and middle-income countries. Genotypic analysis of the observed drug-resistant isolates would have been valuable to verify the findings. Other limitations include that the study did not examine when pregnant women became colonized with ESBL-E. Second, the study did not examine potential risk factors that promote ESBL-E colonization/infection, which would have allowed for any intervention for clinical management. Finally, the study does not document antibiotics taken as self-medication or prescribed during pregnancy.

## 4. Materials and Methods

### 4.1. Ethics

Informed written consent was obtained from all study participants. The study was approved by the Institutional Review Board of Vietnam Military Medical University, Hanoi, Vietnam. The study was conducted between January 2016 and November 2020 at the National Hospital of Obstetrics and Gynecology (NHOG) in Hanoi, Vietnam, the largest healthcare center and a referral hospital for northern Vietnam, serving a population of ~55 million people with 450,000 deliveries annually. Vaginal swab samples (*n* = 3104) were collected from hospitalized pregnant women and screened for Gram-negative bacteria. All clinical and demographic parameters are summarized in Table 1.

### 4.2. Identification and Antimicrobial Susceptibility Testing

Vaginal samples were processed for detection of Enterobacterales according to routine laboratory protocols. In brief, vaginal swab samples were inoculated onto MacConkey agar plates (Merck, Kenilworth, NJ, USA) and incubated at 37 °C for 24 h to grow Gram-negative enteric bacteria. The identified colonies were subjected to direct identification and antimicrobial susceptibility testing using the VITEK^®^ 2 compact automated system (bioMérieux, Lyon, France) with VITEK^®^ 2 GN ID card (bioMérieux, Lyon, France) for species-level identification of clinically important Gram-negative bacilli, including a broad range of Gram-negative Enterobacterales, non-Enterobacterales and other highly pathogenic organisms.

### 4.3. Identification of ESBL-E Phenotypes

ESBL phenotypic confirmation was performed using a combined disc diffusion method recommended by Clinical and Laboratory Standards Institute-2018 (CLSI, Wayne, PA, USA) [39]. Briefly, colonies from the vaginal samples were transferred to 1 mL of normal saline to adjust turbidity to 0.5 McFarland standard. A lawn culture was established with this inoculum on a Mueller–Hinton agar plate. The inoculum was allowed to dry for 15 min, and the antibiotic discs cefotaxime (30 mg)/ceftazidime–clavulanic acid (30 mg/ 10 mg) and ceftazidime (30 mg)/ceftazidime–clavulanic acid (30 mg/ 10 mg) were placed 20 mm apart and incubated for 37 °C for 24 h. The phenotypic test was interpreted according to CLSI guidelines based on the diameter of the zone of inhibition (≥ 5 mm). The quality control of antibiotic susceptibility testing was performed with *E. coli* ATCC 25922 and *K. pneumoniae* ATCC 700603 (for β-lactam/β-lactamase inhibitor combination).

A large proportion of the bacterial isolates were tested for susceptibility to different classes of antibiotics, including β-lactams (ampicillin, amoxicillin/clavulanic acid, ampicillin/sulbactam, piperacillin/tazobactam), cephalosporins/cephamycins (cefazolin, cefepime, cefotaxime, ceftriaxone, cefuroxime, ceftazidime), carbapenems (doripenem, ertapenem, imipenem, meropenem), aminoglycosides (gentamicin, tobramycin, amikacin), tetracycline, quinolones (ciprofloxacin, levofloxacin, norfloxacin), sulfonamides (trimethoprim/sulfamethoxazole), fosfomycin and nitrofurans (nitrofurantoin) using VITEK^®^ 2. In case of unavailability of VITEK^®^ 2 test kits, susceptibility testing was performed by disk diffusion.

### 4.4. Statistical Analysis

Statistical analyses were performed using Stata 13.1 (StataCorp, College Station, TX, USA) and R 4.0.2 [40]. All tests were two-tailed, and a *p*-value of 0.05 was considered statistically significant. Comparisons of associations between Enterobacterales carriage, antibiotic susceptibility were computed using a chi-squared test.

## 5. Conclusions

Pregnant women had a high prevalence of colonization and may transmit ESBL-E to neonates at birth, an important risk factor for early-onset neonatal sepsis. However, further studies are needed to validate our findings and to assess the necessity of ESBL screening in pregnant women. The high prevalence of ESBL-producing Enterobacterales and carbapenem resistance in *Klebsiella* spp. in Vietnam is alarming and highlights the necessity for implementing AMR surveillance in Vietnamese hospitals. Most importantly, access to molecular typing methods in low- and middle-income countries is desperately needed to better understand spreading AMR in the global context.

## Figures and Tables

**Table 1 antibiotics-10-00572-t001:** Baseline characteristics of enrolled participants.

Characteristics	*n* (%)
Enrolled	3863
Sample collected	3104
Positive for Enterobacterales	918 (30)
Mean age (range)	30 (17–46)
Gestational age in weeks (range)	31 (16–41)
918 individuals (positive isolates)	
High risk of preterm birth	150 (16)
Stillbirth	23 (3)
Primiparous	284 (31)
2nd Parity	244 (27)
3rd Parity	119 (13)
4th Parity	24 (3)
5th Parity	4 (0.4)
6th Parity	1 (0.1)
Parity (not available)	242 (26)
In vitro fertilization (IVF)	162 (18)
Occupation of 918 individuals (positive isolates)	
Self-employed	281 (31)
Army personal	1 (0.1)
Teacher	94 (10)
Officer	197 (21)
Worker	183 (20)
Farmer	96 (10)
Student	3 (0.3)
Healthcare worker	34 (4)
Engineer	5 (0.5)
Housewife	24 (3)

**Table 2 antibiotics-10-00572-t002:** Enterobacterales carriage among 3104 samples from pregnant women.

	Total (%), *n* = 3104	ESBL-Positive *n* (%)	ESBL-Negative *n* (%)	*p*-Value
*E. coli*	667 (21)	340 (11)	327 (11)	NS
*K. pneumoniae*	243 (8)	88 (3)	155 (6)	NS
*K. oxytoca*	8 (0.3)	4 (0.1)	4 (0.1)	NS
	918 (30)	432 (14)	486 (16)	NS

NS: non-significant; ESBL = extended-spectrum beta-lactamase.

**Table 3 antibiotics-10-00572-t003:** Phenotypic antibiotic resistance of Enterobacterales from vaginal swabs of Vietnamese pregnant women 2016–2020 (*n* = 904).

		All Isolates ^a^					*Escherichia coli* ^a^				*Klebsiella* spp. ^a^		
Group	Antimicrobial Agents	Total, *n* = 904 ^a^	ESBL^+^, *n* = 422	ESBL^-^, *n* = 482	*p*-Value	Total, *n* = 657 ^a^	ESBL^+^, *n* = 332	ESBL^-^, *n* = 325	*p*-Value	Total, *n* = 247 ^a^	ESBL^+^, *n* = 90	ESBL^-^, *n* = 157	*p*-Value
		% (*n*)	% (*n*)	% (*n*)		% (*n*)	% (*n*)	% (*n*)		% (*n*)	% (*n*)	% (*n*)	
Penicillins	Ampicillin	94 (852)	100 (422)	89 (430)	<0.001	92 (605)	100 (332)	84 (373)	<0.001	100 (247)	100 (90)	100 (157)	<0.001
Β-lactam combination	Ampicillin/sulbactam ^b^	73 ^b^ (507)	77 ^b^ (236)	70 ^b^ (271)	0.04	74 ^b^ (373)	73 ^b^ (181)	76 ^b^ (192)	0.5	71 ^b^ (134)	96 ^b^ (55)	60 ^b^ (79)	0.5
Cephalosporin	Cefazolin ^c^	63 ^c^ (431)	99 ^c^ (302)	34 ^c^ (129)	<0.001	62 ^c^ (307)	99 ^c^ (245)	25 ^c^ (62)	<0.001	66 ^c^ (124)	100 ^c^ (57)	51 ^c^ (67)	<0.001
	Ceftriaxone	56 (502)	99 (418)	17 (84)	<0.001	56 (368)	99 (329)	12 (39)	<0.001	54 (134)	99 (89)	29 (45)	<0.001
	Cefepime ^d^	49 ^d^ (422)	88 ^d^ (361)	13 ^d^ (61)	<0.001	49 ^d^ (313)	89 ^d^ (290)	7 ^d^ (23)	<0.001	50 ^d^ (109)	87 ^d^ (71)	28 ^d^ (38)	<0.001
Carbapenems	Ertapenem	8 (72)	1 (5)	14 (67)	<0.001	3 (22)	1 (2)	6 (20)	<0.001	20 (50)	3 (3)	30 (47)	<0.001
	Imipenem ^e^	9 ^e^ (55)	2 ^e^ (6)	15 ^e^ (49)	<0.001	4 ^e^ (16)	0 ^e^ (0)	7 ^e^ (16)	<0.001	23 ^e^ (39)	9 ^e^ (6)	33 ^e^ (33)	<0.001
Aminoglycosides	Gentamicin	41 (375)	50 (211)	34 (164)	<0.001	42 (276)	47 (157)	37 (119)	0.006	40 (99)	47 (157)	60 (54)	<0.001
Quinolones	Ciprofloxacin	41 (372)	56 (237)	28 (135)	<0.001	46 (301)	62 (206)	29 (95)	<0.001	29 (71)	34 (31)	25 (40)	0.1
Sulfonamides	Trimethoprim/sulfamethoxazole	69 (625)	77 (327)	62 (298)	<0.001	75 (493)	81 (269)	69 (224)	<0.001	53 (132)	64 (58)	47 (74)	0.009
Nitrofurans	Nitrofurantoin ^f^	24 ^f^ (178)	22 ^f^ (72)	26 ^f^ (106)	0.2	4 ^f^ (24)	7 ^f^ (17)	3 ^f^ (7)	0.03	77 ^f^ (154)	83 ^f^ (55)	73 ^f^ (99)	0.1
MDR ^g^	≥ 3 antibiotic classes	51 (463)	65 (275)	39 (188)	<0.001	56 (369)	70 (231)	42 (138)	0.001	38 (94)	49 (44)	32 (50)	0.008

Abbreviations: MDR = multidrug resistant; ESBL = extended-spectrum beta-lactamase; ESBL^+^ = ESBL-positive; ESBL^-^ = ESBL-negative ^a^ only isolates with complete antimicrobial susceptibility results for ampicillin, ceftriaxone, ertapenem, gentamicin, ciprofloxacin and trimethoprim/sulfamethoxazole (*n* = 904/918) and ≥50% tested isolates are included.; ^b^ ampicillin/sulbactam susceptibility data not available for 214 isolates (*E. coli*, *n* = 156 and *Klebsiella* spp., *n* = 58); ^c^ cefazolin susceptibility data not available for 216 isolates (*E. coli*, *n* = 158 and *Klebsiella* spp., *n* = 58); ^d^ cefepime susceptibility data not available for 42 isolates (*E. coli*, *n* = 14 and *Klebsiella* spp., *n* = 28); ^e^ imipenem susceptibility data not available for 314 isolates (*E. coli*, *n* = 235 and *Klebsiella* spp., *n* = 79); ^f^ nitrofurantoin susceptibility data not available for 168 isolates (*E. coli*, *n* = 122 and *Klebsiella* spp., *n* = 46), ^g^ multi-drug resistance (MDR) is defined as resistance to three of more classes of antibiotics. All beta-lactams were considered as one class.

## Data Availability

The data supporting reported results are available on request.

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
