# Peer review of "Maternal Vaginal Colonization and Extended-Spectrum Beta-Lactamase-Producing Bacteria in Vietnamese Pregnant Women"

_antibiotics, 2021, doi:10.3390/antibiotics10050572_

Round 1

Reviewer 1 Report

The authors improved the manuscript. Nonetheless, and despite the fact I understand the authors explanation, I think this study is very incomplete without genotypic data. Most of the papers published nowadays include whole genome sequencing, while this study is purely phenotypic.

Author Response

We apologize, if we have not delivered adequate answers to your comments/questions. We have elaborated on your comments from R1, R2 and R3. 

Unfortunately, since the data was generated in the routine microbiological diagnostics, priority was given to the phenotypic susceptibility testing, since this is more relevant to patient care than molecular typing data, which would be interesting for surveillance and scientific purposes. Retrospective analysis/characterization was not possible, since the isolates were not cryo-preserved. We also agree that we should co-operate with other laboratories/groups, who have access to molecular typing. Our data indicated that AMR is a problem, and that surveillance and molecular typing is essential. Despite the lack of molecular typing, we hope that our findings will increase awareness of the AMR problematic in LMIC (such as SE-Asia). We have initiated future collaborations to include high-resolution molecular typing method for subsequent studies. Unfortunately, since samples from this study were no longer available, molecular typing could not be performed. We have included this in the limitations.

Reviewer 2 Report

I do not think the authors have respond to the feedback. The study lacks the genotypic analysis of multidrug resistant isolates to verify the data.

Author Response

Yes, we agree that genotypic characterization would be valuable for confirmation and also to get an insight on the molecular epidemiology and spread of CTX-M genes. However, since the presented data was acquired through routine microbiological diagnostics, it was not possible to perform routine molecular characterization due to financial and personnel restrictions in this hospital setting. Routine molecular typing would have taken up too many resources, which was needed for routine diagnostics and since molecular typing does not have any direct consequences to patient care, phenotypic testing was prioritized. Nonetheless, we believe that our findings contribute some valuable epidemiological data to increase awareness of emerging/on-going AMR problems in South-East Asia and especially in low to middle income countries. In addition, our data also emphasizes the necessity of surveillance and access to molecular typing (PCR or even WGS) for low- and middle-income countries, which may encourage some of the high-income countries with access to these technologies to co-operate with low- and middle-income countries.

Unfortunately, since the bacterial isolates were not frozen / cryo-preserved, a retrospective molecular typing is not possible.

We have added this aspect as a limitation in the discussion. 

Reviewer 3 Report

This study investigated the prevalence of ESBL-producing Enterobacteriaceae in the birth-canal of 3104 Vietnamese pregnant women between 2016-2020 via vaginal swabs. The high rate of ESBL-producers and carbapenem resistance in Enterobacteriaceae in Vietnam is an important risk factor for ESBL-E transmission to neonates at birth, emphasizes the need for consequent surveillance and access to molecular typing.

The manuscript is well presented. Just a few minor points:

Line 37: Abstract. Need a space in between "Spp.Pregnant"

Line 91: To be clearer, need to add "in all swab samples" before "were 14% (n=432)" Please add "ESBL-production was detected in 47% (432/918) of Enterobacteriaceae." after "16% (n= 486) respectively."

Line 128: need a space in between "swabsof" in the Table 3 heading.

Line 198: "There are few study limitations" should be "a few", missing "a".

Line 215-216: Need to add ethics approval number. 

Author Response

Response to Reviewer #3

This study investigated the prevalence of ESBL-producing Enterobacteriaceae in the birth-canal of 3104 Vietnamese pregnant women between 2016-2020 via vaginal swabs. The high rate of ESBL-producers and carbapenem resistance in Enterobacteriaceae in Vietnam is an important risk factor for ESBL-E transmission to neonates at birth, emphasizes the need for consequent surveillance and access to molecular typing.

The manuscript is well presented. Just a few minor points:

>>Response: Thank you for the positive feedback and your suggestions, we have made the suggested changes in the revised manuscript.

Line 37: Abstract. Need a space in between "Spp.Pregnant"

>>Response: done

Line 91: To be clearer, need to add "in all swab samples" before "were 14% (n=432)" Please add "ESBL-production was detected in 47% (432/918) of Enterobacteriaceae." after "16% (n= 486) respectively."

>>Response: Thank you for this suggestion. We agree that these changes would make the data presentation clearer and have made these changes in the revised manuscript.

Line 128: need a space in between "swabsof" in the Table 3 heading.

>>Response: done

Line 198: "There are few study limitations" should be "a few", missing "a".

>>Response: done. Thanks for pointing this out.

Line 215-216: Need to add ethics approval number.

 >>Response: The study was conducted according to the guidelines of the Declaration of Helsinki, and approved by the Institutional Review Board of Vietnamese Military Medical University, Hanoi, Vietnam (VMMU-IEC-AMR- 02-20201603-V2). Added in the Ethical Approval

Reviewer 4 Report

Classically, the screening of pregnant microbial risk environment  in the vagina is characterised first by GAS - Streptococcus beta-hemolytic group A screening. Prenatal detection of GAS colonisation is recommended to prevent GAS neonatal septicemia. While it is known that early neonatal sepsis (preculiarly in premature neonates) can also occur due to Escherichia coli or other Gram-negative bacillae, there is no standard recommendation to screen for these germs. The main question is to elucidate the objective of such a screening study in pregnant women: could you please give more information on your objective in the introduction? It looks as if you have performed a screening of Gram-negative bacteria in an unusual site of sampling (vagina): why?  

As you know, screening recommendations for multidrug-resistant Gram-negative bacteria comprise microbiological analyses from rectal swabs. Why do you choose now vaginal samples in pregnant women?

Do you screen in parallel for GAS - Streptococcus beta-hemolytic A? If yes, why don't you present these results in parallel with those of Gram-negative bacteria?

The logical steps for justifying your study are not evident.

Minor remark: ESBL+ve and ESBL-ve : what does "ve" mean?

Author Response

Response to Reviewer #4

Classically, the screening of pregnant microbial risk environment  in the vagina is characterised first by GAS - Streptococcus beta-hemolytic group A screening. Prenatal detection of GAS colonisation is recommended to prevent GAS neonatal septicemia. While it is known that early neonatal sepsis (preculiarly in premature neonates) can also occur due to Escherichia coli or other Gram-negative bacillae, there is no standard recommendation to screen for these germs. The main question is to elucidate the objective of such a screening study in pregnant women: could you please give more information on your objective in the introduction? It looks as if you have performed a screening of Gram-negative bacteria in an unusual site of sampling (vagina): why?

>>Response: Group B Streptococcus (GBS) and Enterobacteriaceae are independent risk-factors for the acquisition of neonatal sepsis. Group A streptococcus is NOT a risk factor for neonatal sepsis. The reviewer might have meant GBS, instead of GAS (GAS can cause other symptoms such as vaginitis, etc. but it’s not relevant in this context). We agree that GBS is an established risk factor for neonatal sepsis but the treatment of GBS-related infections are not problematic. The emergence and acquisition of resistance in streprococcus spp other than Streptococcus pneumonia is not a major problem. Most if not all GBS (S. agalactiae) are susceptible to penicillins (e.g. PMID: 25031448). Furthermore, there are prophylactic measures to prevent GBS early onset sepsis by administering antibiotics pre-delivery. Frankly, we do not see any added scientific value to add GBS in the presented manuscript, since it blurs the focus on the emergence of drug-resistant Gram-negatives (Enterobacteriaceae), which is a major and global problem. Recently, ESBL vaginal colonization has been demonstrated as a risk factor for vertical transmission of ESBL during delivery. This is why we investigated the prevalence of ESBL in vaginal swabs of pregnant women. We have revised the introduction to clarify the objectives of the study, as suggested by the reviewer.

As you know, screening recommendations for multidrug-resistant Gram-negative bacteria comprise microbiological analyses from rectal swabs. Why do you choose now vaginal samples in pregnant women?

#>>Response: Recent reports demonstrated the relevance of vaginal colonization with ESBL-producing bacteria as an independent risk-factor for neonatal sepsis (PMID: 31991333, PMID: 23515552, PMID: 33673648). Of course cross-contamination between rectal and vaginal colonization cannot be ruled out. Considering the evidence from observational and cohort studies, we believe that vaginal colonization is more relevant in this case, since the exposition to bacteria from the vagina and birth canal is more significant than intestinal colonization as babies are usually not delivered via the rectum. Rectal swabs can be used if you are only interested in the carriage prevalence but rectal swabs DO NOT have any therapeutical/clinical relevance in this context.

Do you screen in parallel for GAS - Streptococcus beta-hemolytic A? If yes, why don't you present these results in parallel with those of Gram-negative bacteria?

>>Response: GBS is not the focus of this study. GBS as a risk factor is known and as we have mentioned, this would blur the main findings and is not relevant in terms of AMR or emergence of resistance.

The logical steps for justifying your study are not evident.

>>Response: As mentioned, besides GBS, E. coli and other Gram-negatives are one of the main causes of neonatal sepsis. One of the most effective ways to prevent neonatal sepsis is by administering antibiotic prophylaxis (such as penicillin for GBS) prior to delivery. For Gram-negatives, drug-resistance is emerging globally. In many regions (such as South and South-East Asia) the rate of drug-resistant Enterobacteriaceae is high but epidemiological data is lacking. Theferore, we performed this study to get an estimate of the prevalence of ESBL carriage in pregnant women in Vietnam. Resistance prevalence data is useful for calculated empirical therapy. Since cephalosporines are one of the highly used antibiotics (and the safer choice for use in pregnancy), we think that our study and our approach is justified. We have added a few paragraphs in the introduction to emphasize and to clarify this point.

Minor remark: ESBL+ve and ESBL-ve : what does "ve" mean?

>>Response: ESBL+ve=ESBL positive and ESBL-ve=ESBL-negative. We have added this to the abbreviations in the footnote of Table 3 and have changed it to ESBL+  ESBL- for clarity.

Round 2

Reviewer 4 Report

Thank you for your improvements and sorry for the unforgivable confusion GAS/GBS (besides midnight tiredness) .

I understand better your approach: by analogy to GBS, which is also present in intestinal material AND in vaginal cavity, you opted for a microbiological survey on vaginal samples - which is not usual for enterobacteriacae. The two approaches (vaginal sample or rectal sample) would likely give similar results of descriptive epidemiolgy. But you are obviously right to consider that the direct relationship of vaginal flora with early onset septicemia in neonates could be likely stronger than with rectal samples. So, your alternative approach deserves interest, and maybe opens a way to revised procedure. Now, for the future, even for GBS, there is no consensus on benefit of systematic screening of prenatal bacterial portage (see recent Cochrane review on GBS screening during pregnancy). Your manuscript has an interest on methodolocial descriptive methodology, but remain very careful: there is far way from this interesting observation to the decision to screen pregnant women for enterobacteriacae (with or without AMR). I recognize that you do not support this perspective neither in the present manuscript.

Author Response

Thank you for your helpful suggestions. We are not sure, if we understood your comme. The conclusion was carefully written to not make any suggestions or to imply that screening of Enterobacterales or ESBL should be implemented on the basis of our findings. We merely suggested that (in general) AMR might be problematic in Vietnam, as suggested by the high prevalence of MDR in our findings and this did not refer to maternal or newborn screening procedures. Therefore we think that this conclusion is valid. Nonetheless, we have added: (Line 287) However, further studies are needed to validate our findings and to assess the necessity of ESBL screening in pregnant women.